



# Mind-the-gap part I: Accurately locating warm marine boundary layer clouds and precipitation using spaceborne radars

Katia Lamer[1,2**], Pavlos Kollias[2,3,4], Alessandro Battaglia[5,6] and Simon Preval[5]

[1] City University of New York affiliation
[2] Brookhaven National Laboratory
[3] Stony Brook University
[4] Cologne University
[5] University of Leicester, Leicester, UK
[6] UK National Centre for Earth Observation
[*] Affiliation when work was conducted
[**] Current affiliation

*Correspondence to:* Katia Lamer (klamer@bnl.gov)

**Abstract**
Ground-based radar observations show that, in the eastern north Atlantic, 50% of warm marine boundary layer
(WMBL) hydrometeors occur below 1.2km and have reflectivities < -17dBZ, thus making their detection from space
susceptible to the extent of surface clutter and radar sensitivity.
Surface clutter limits the CloudSat-Cloud Precipitation Radar (CPR)'s ability to observe true cloud base in ~52% of
the cloudy columns it detects and true virga base in ~80%, meaning the CloudSat-CPR often provides an incomplete
view of even the clouds it does detect. Using forward-simulations, we determine that a 250-m resolution radar would
most accurately capture the boundaries of WMBL clouds and precipitation; That being said, because of sensitivity
limitations, such a radar would suffer from cloud cover biases similar to those of the CloudSat-CPR.
Overpass observations and forward-simulations indicate that the CloudSat-CPR fails to detect 29-41% of the cloudy
columns detected by the ground-based sensors. Out of all configurations tested, the 7 dB more sensitive EarthCARE-
CPR performs best (only missing 9.0% of cloudy columns) indicating that improving radar sensitivity is more
important than shortening surface clutter for observing cloud cover. However, because 50% of WMBL systems are
thinner than 400 m, they tend to be artificially stretched by long sensitive radar pulses; hence the EarthCARE-CPR
overestimation of cloud top height and hydrometeor fraction.
Thus, it is recommended that the next generation of space-borne radars targeting WMBL science shall operate
interlaced pulse modes including both a highly sensitive long-pulse and a less sensitive but clutter limiting short-pulse
mode.





## 1 Introduction

Because of their ubiquitous nature and of the way they interact with solar and longwave radiation, warm marine boundary layer (WMBL) clouds play a crucial role in the global energy budget [*Klein and Hartmann*, 1993]. Unfortunately, numerical models still struggle to properly represent their coverage, vertical distribution, and brightness (e.g., [*Nam et al.*, 2012]). This uncertainty ultimately affects our confidence in future climate projections [*Bony et al.*, 2015; *Sherwood et al.*, 2014]. Climate simulations could be improved from comparisons with additional observations of the macrophysical and microphysical properties of WMBL clouds, as well as from improvements in our understanding of the relationships between low-level clouds and their environment.

Millimeter-wavelength radar signals, because of their ability to penetrate clouds, have long been used to document the vertical distribution of WMBL clouds (e.g., [*Haynes et al.*, 2011; *Sassen and Wang*, 2008]) and their internal structure (e.g., [*Bretherton et al.*, 2010; *Dong and Mace*, 2003; *Huang et al.*, 2012; *Lamer et al.*, 2015]) as well as to identify precipitation (e.g., [*Ellis et al.*, 2009; *Leon et al.*, 2008; *Rapp et al.*, 2013]) and characterize its vertical structure (e.g., [*Burleyson et al.*, 2013; *Comstock et al.*, 2005; *Frisch et al.*, 1995; *Kollias et al.*, 2011]). However, the representativeness of radar observations largely depends on factors such as coverage, radar sensitivity, vertical/horizontal resolution and on the presence of clutter.

Spaceborne radars are often preferred over ground-based and airborne ones because of their ability to cover vast areas of the globe [*Battaglia et al.*, Submitted]. The first spaceborne Cloud Precipitation Radar (CPR) designed to detail the vertical structure of clouds was launched in 2006 onboard CloudSat [*Stephens et al.*, 2002]. The CloudSat-CPR is still operational; it transmits a 3.3 microsecond pulse with a 1.4 km field of view at the surface and can achieve a sensitivity of -28 dBZ after its measurements are averaged in 0.32-s time intervals and sampled at 0.16-s along its nadir track [*Stephens et al.*, 2002]. However, the CloudSat-CPR's long power pulse also generates a surface clutter echo which tends to partially mask signals from cloud and precipitation forming below circa 1 km [*Marchand et al.*, 2008]. For this reason, the CloudSat-CPR's actual ability to document WMBL clouds and precipitation remains uncertain.

Comparison of various satellite-based cloud products suggest that globally the CloudSat-CPR can only detects roughly 30-50% of all WMBL cloud-containing atmospheric columns [*Christensen et al.*, 2013; *Liu et al.*, 2018; *Liu et al.*, 2016; *Rapp et al.*, 2013]. According to *Christensen et al.* [2013] most of the CloudSat-CPR cloud cover bias is due to its inability to detect clouds forming entirely within the region occupied by its surface clutter. *Rapp et al.* [2013] instead attribute this deficiency mainly to the CloudSat-CPR's sensitivity which they believe is insufficient to detect the small droplets composing WMBL clouds like those forming in the southeastern Pacific region. However, in another study, *Liu et al.* [2018] concluded that the coarse resolution of the CloudSat-CPR has more of an impact on its ability to detect all cloudy columns than surface clutter and limited sensitivity. Such a lack of consensus makes designing more effective radar architectures for future spaceborne missions more complicated. Also, because most existing CloudSat-CPR-performance assessments are based on observations from (visible) sensors that cannot





penetrate cloud top, there is little to no information about the CloudSat-CPR's ability to holistically document the vertical structure of those cloudy columns it detects (i.e., provide information from cloud top to cloud base and of virga and rain below cloud).

It is not uncommon to rely on observations collected by highly sensitive airborne and ground-based millimeter radar observations to assess the performance of coarser less sensitive radars (e.g., [*Burns et al.*, 2016; *Lamer and Kollias*, 2015]). Such observations have allowed *Stephens et al.* [2002] to conclude that, based-on sensitivity alone, the CloudSat-CPR should only be able to detect 70% of marine boundary layer cloud segments. A study considering the impact of the CloudSat-CPR's rather coarse vertical resolution, large horizontal field of view and surface clutter would complement this preliminary work and allow for a more rigorous quantification of its ability to document the vertical distribution of cloud fraction.

Instrument geometry effects are best accounted for in forward simulators. Using ground-based observations and an instrument forward-simulator *Burns et al.* [2016] determined that the CloudSat-CPR's successor, the EarthCARE-CPR [*Illingworth et al.*, 2015], will only detect 70-80% of marine boundary layer cloud segments; moreover its coarse vertical resolution (500 m, same as the CloudSat-CPR) will introduce significant biases in reported cloud boundaries. These results however likely need be revised since changes have since been made to the design of this joint European Space Agency (ESA) and Japanese Aerospace Exploration Agency (JAXA) spaceborne mission (https://earth.esa.int/web/guest/missions/esa-future-missions/earthcare).

Along those lines, the current study relies on the use of instrument forward simulators and on observations collected by the ground-based Ka-band ARM Zenith radar (KAZR) and the ceilometer operating at the Atmospheric Radiation Measurements (ARM) program Eastern North Atlantic (ENA) facility to document the properties of WMBL clouds and precipitation with the goal of:

- o  quantifying the CloudSat-CPR's ability to estimate their coverage and vertical distribution as well as its accuracy in determining the location of cloud tops and cloud/virga base (Sect. 3.0);

- o  identifying which property (thickness, reflectivity, vertical location) of WMBL clouds and precipitation mostly complicate their detection from space (Sect. 4.0);

- o  evaluating the performance of alternative radar configurations designed for an optimum characterization of WMBL clouds and precipitation (Sect. 5.0).

## 2    Datasets

This study focuses on evaluating how well spaceborne CPR are able to document the properties of warm marine boundary layer (WMBL) clouds. We define WMBL clouds as cloudy columns with the highest cloud top below 5.5



km/500 mb and warmer than 0°C. This definition limits our analysis to WMBL regimes not associated with mid- or
high- clouds aloft but does not exclude periods where multiple WMBL cloud layers overlap.

The next sub-sections describe how we extracted cloud and precipitation information from raw CloudSat-CPR – to
evaluate the performance of current spaceborne sensors in this regime – (Sect. 2.1), ARM measurements – which act
as a benchmark – (Sect. 2.2) and how we forward-simulate alternative spaceborne radar configurations (Sect. 2.3).

**2.1    CloudSat Spaceborne W-band Radar Observations**

The CloudSat-CPR has been collecting observations since May 2006; Initially twice a day, but then only once a day
(at 15:00 UTC) after it returned to the A-Train in May 2012 following a spacecraft battery failure [*Stephens et al.*,
2018]. Periods when CloudSat passed within a 200 km radius of the ARM ENA ground-based facility are used to
evaluate the CloudSat-CPR's ability to characterize WMBL clouds and precipitation (results presented in Sect. 3.0);
this happened 117 times since the ground-based site was made permanent at the end of 2015 (daytime only). The
GEOPROF granules (algorithm version 4.0) corresponding to these overpasses were identified and extracted for
analysis following the method of *Protat et al.* [2009]. Variables taken from this product include Radar_Reflectivity,
CPR_Cloud_mask (hydrometeor echo mask), and CPR_Echo_Top (cloud type classification). An example of raw
radar reflectivity observations collected by the CloudSat-CPR on February 27, 2016 is given in Fig. 1c.

The GEOPROF product provides observations sampled every ~240 m in range and ~1.0 km along-track taken from
the CloudSat-CPR native 500-m range resolution and ~1.7km along-track by 1.3km across-track field of view
[*Stephens et al.*, 2002; *Tanelli et al.*, 2008]. The CloudSat-CPR's raw radar reflectivity measurements are filtered for
clutter and noise using the CPR_Cloud_mask. Progressively more aggressive masks are applied until a compromise
is reached between the number of detectable hydrometeors and the amount of remaining noise. Radar reflectivities are
first masked for bad and missing echoes (mask value -9; Fig. 1d), then for echoes with significant return power likely
affected by - or resulting from- surface clutter (mask value 5; Fig. 1e). Comparison of Fig. 1d and 1e illustrate that a
majority of the hydrometeor echoes with significant return power are deemed affected by the surface clutter echo and
that following their removal the CloudSat-CPR's ability to detect clouds and precipitation appears significantly
reduced. Since further removing echoes labeled as very weak (mask value 6-20) helps clean up the remaining radar
reflectivity time-height image while minimally affecting the number of detected hydrometeor echoes, our evaluation
of the CloudSat-CPR's performance is based only on echoes deemed weak to strong (mask value >= 20; Fig. 1f).
According to estimates by *Marchand et al.* [2008] these echoes should have less than a 5% chance of being false
hydrometeor detections.

WMBL clouds are isolated using the CPR_Echo_Top mask; Profile with high clouds (mask value 2), mid-level clouds
(mask value 3) and multi-layer clouds (mask value 5) are filtered out leaving low-level clouds, clear, and undetermined
profiles (mask values 4, 1 and 0 respectively; Fig. 1b). We additionally filter out profiles that have their maximum
reflectivity more than 150 m away from 0 m height; this last step is intended to identify profiles for which the CloudSat-


CPR was mispointing, which leads to vertical offset in the surface peak return.

**2.2     ARM Ground-based Observations**

The ARM program's KAZR is a 34.86 GHz (i.e., Ka-band) radar able of generating a 4 microsecond long symmetrical
vertical pulse creating a 0.3° wide 3-dB beamwidth. Following signal integration (1-s, 6,000-pulses), this radar
achieves a -44 dBZ minimum detectable signal (MDS) at 1 km. The KAZR is able to collect observations from 87 m
above ground to 18 km at ~30 m vertical resolution and 2 s time resolution [*Lamer et al.*, 2019]. Because the KAZR's
observations are not oversampled in the vertical, they are considered more independent than that of the CloudSat-
CPR.

We analyze the complete data record collected by the ground-based ARM sensors between October 2015 and
November 2017 (719 days) to 1) characterize the properties of WMBL clouds and precipitation (results in Sect. 4,0)
and 2) to evaluate the performance of theoretical radar architectures in detecting those clouds (results in Sect. 5.0).
This period also includes the 117 CloudSat overpass days, which we analyze separately to identify gaps specific to
the currently deployed CloudSat-CPR (results in Sect. 3.0).

For each analysis, we extract several complementary datasets from the ARM archive: i) KAZR general mode
(processing level a1): reflectivity, snr_copol (co-polar signal to noise ratio), ii) ceilometer: first_cloud_base_height,
iii) Parsivel laser disdrometer: equivalent radar reflectivity, and iv) radiosonde: temperature.

KAZR signal-to-noise ratio measurements are used as input to the *Hildebrand and Sekhon* [1974] algorithm to
distinguish significant echoes (hydrometeors and clutter) from noise. Liquid cloud base height determination from
collocated ceilometer is used to isolate radar echoes associated with cloud (above the first liquid cloud base height)
and precipitation (below the first liquid cloud base height) and to filter out clutter in the subcloud layer. Clutter filtering
is based on the argument that precipitation falling from cloud base should be continuous, thus any echo in the subcloud
layer detached from the main echo is labelled as clutter and is filtered out. All echoes thinner than 90m (3 range gates)
are also labelled as clutter and filtered out; comparison with the ceilometer confirms that this step does lead to the
removal of cloudy echoes. An example of processed radar reflectivity from KAZR is depicted in Fig. 1a.

Filtered KAZR radar reflectivity measurements are corrected for gas attenuation following *Rosenkranz* [1998] and
calibrated using observations collected during light precipitation events by the collocated surface-based Parsivel laser
disdrometer as well as using observations from the CloudSat-CPR collected over a small radius around the site
following *Kollias et al.* [2019].

WMBL cloud profiles are isolated from ice and high cloud containing profiles using KAZR radar reflectivity and
sonde temperature information. Only profiles having echoes below 5.5 km or below the height of the 0°C isotherm,
whichever one is lowest, are considered in this analysis.





**2.3   Forward-simulations based on ground-based KAZR observations**

Forward simulations are conducted to improve our understanding the CloudSat-CPR limitations and to identify possible modifications which could lead to improvements in the detection of WMBL clouds (results in Sect. 5.0). We forward simulate seven radar architectures. The first four are based on the CloudSat-CPR's current configuration gradually improving each of its capabilities until it matches the configuration of the EarthCARE-CPR. The EarthCARE-CPR design includes several improvements over CloudSat, namely:

1)   a new asymmetrical point target response,

2)   enhanced sensitivity,

3)   a smaller field of view and integration distance, and

4)   increased range oversampling.

The EarthCARE-CPR will also be the first spaceborne atmospheric radar capable of documenting the movement of hydrometeors. This capability has been evaluated in several publications such as *Schutgens* [2008], *Battaglia et al.* [2013] , *Kollias et al.* [2014], *Sy et al.* [2014], and *Burns et al.* [2016] and is beyond the scope of this study. The last two architectures are based on propositions made in the context of the National Aeronautics and Space Administration (NASA)'s future Aerosol and Cloud, Convection and Precipitation (ACCP) mission (https://science.nasa.gov/earth-science/decadal-accp). They both have:

1)   increased range resolution but,

2)   reduced sensitivity

Specifications for each radar configuration are given in Table 1 and Fig. 2.

Processed (i.e., filtered, corrected and calibrated) KAZR radar reflectivity observations (time-height) are used as input to the forward-simulations. First, assuming a constant horizontal wind speed of 10 m s$^{-1}$, the KAZR time axis is converted to horizontal distance. Then, to emulate the surface reflectivity which is not seen by KAZR, an artificial surface echo is added to the processed KAZR reflectivity field at 0 m altitude (see Appendix I for more information on how real CloudSat-CPR observations were used to construct this surface echo). Each spaceborne radar configuration is simulated by first horizontally convolving the high-resolution (30 m x 20 m) KAZR reflectivity fields using an along-track weighting function represented using a symmetrical gaussian distribution covering a distance equivalent to 2 times the along-track field of view and then by vertically convolving the horizontally convolved reflectivity field using either of the two range-weighting functions depicted in Fig. 2. The asymmetrical range weighting function is modelled after that of the EarthCARE-CPR which was obtained from prelaunch testing of the EarthCARE-CPR (provided by the mission's engineering team). The symmetrical range-weighting function used (only) for the CloudSat$_f$ forward simulation is modelled using a gaussian distribution adjusted to produce a surface





clutter echo profile similar to that observed by the CloudSat-CPR post-launch (more information in Appendix I).
Finally, along-track integration is emulated by averaging the convolved profiles in sections dictated by the integration
distance of each spaceborne radar without overlap between the section. Note that these forward-simulations are two
dimensional and as such do not capture cross-track effects; Also note that liquid attenuation and noise are not
represented.
For cloud and precipitation characterization, the forward-simulated radar reflectivity fields are finally filtered for
surface clutter. To do this, forward simulations of clear sky conditions are used to estimate the vertical extent and
intensity of surface clutter. For each radar configuration, for all heights affected by surface clutter, the clear sky surface
clutter reflectivity is removed from the forward-simulated radar reflectivity and only echoes with reflectivity at least
3 dB above the surface clutter reflectivity are conserved and deemed reliable. Otherwise, for all heights above the
surface clutter, only those echoes with reflectivity below the radar MDS are filtered out.

**2.4    Evaluation metrics**

Radars alone do not have the capability to distinguish between clouds and precipitation. For this reason, we often refer
to them as hydrometeor layers. The current study aims at characterizing:

i)    the base of the lowest hydrometeor layer (cloud or virga base being indistinguishable), which we take to

be the height of the lowest radar echo in the profile;

ii)    the top of the highest hydrometeor layer (i.e. cloud top), which we take to be the height of the highest

radar echo in the profile;

iii)   the depth covered by hydrometeor layers, which we estimate as the distance between the top of the

highest hydrometeor layer and the base of the lowest hydrometeor layer.

Note that we report hydrometeor boundary heights at the center point of each radar's vertical range gate and not as its
upper or lower limit. This distinction, while seemingly insignificant for radars operating at a fine range sampling (e.g.,
KAZR 30 m), can become important for radar systems having a coarse range sampling (e.g., the CloudSat-CPR 240
m).

We also estimate over the entire observation periods:

i)    hydrometeor cover, defined as the sum of all profiles containing at least one boundary-layer hydrometeor

echo divided by the total number of observed profiles (excluding those determined to contain high, deep

or ice clouds);

ii)    the hydrometeor fraction profile, which we take is the number of boundary-layer hydrometeor echo at

each height divided by the total number of observed profiles (excluding those determined to contain

high, deep or ice clouds).



## 3    Gaps

Figure 1 illustrates examples of observations collected on Feb 27, 2016 near the ENA observatory. The ground-based KAZR radar and ceilometer detected the presence of a thin (up to ~270 m) cloud layer whose properties varied throughout the day. Between 0:00 and 10:00, cloud top height was observed to rise at a rate of roughly 21m hr$^{-1}$. Shortly after 10:00, the KAZR detected signs of drizzle below the ceilometer-detected cloud base height at 941 m. The vertical extent of this drizzle was observed to increase over the course of the day, until it eventually reached 87 m altitude (the lowest altitude at which KAZR measures) around 20:00. Besides changes in cloud top and hydrometeor layer base height, the KAZR also measured changes in the radar reflectivity over the course of the day with more intense radar reflectivity recorded coincidently with deeper drizzle shafts.

At 15:05, CloudSat overpassed within 200 km of the KAZR and ceilometer location. Although the subset of noise-and-clutter-filtered CloudSat-CPR observations show the presence of a hydrometeor layer , the hydrometeor layer detected by the CloudSat-CPR had both breaks, a higher top (1.28 vs. 1.07 km) and a higher base (1.15 vs. 0.51 km) than that detected by KAZR misleadingly making it appear thinner overall (Fig. 1b).

To illustrate how the aforementioned example is representative of the general picture of the WMBL cloud regimes at the ENA, we also compared statistics of hydrometeor layer properties estimated for 89 of the 117 days where CloudSat overpassed within 200 km of the ENA and boundary-layer clouds were the dominant cloud type (Fig. 3 and 4). For this comparison, only KAZR and ceilometer observations taken within 4 hrs of the overpass are considered.

First, agreement between the KAZR reported cloud cover and the ceilometer reported cloud cover confirms that the KAZR's sensitivity is sufficient to detect even the most tenuous clouds forming in this marine boundary layer regime; this makes the KAZR an ideal sensor to document the properties of WMBL clouds and evaluate the CloudSat-CPR's performance (Fig. 3a). Although not expected to perfectly match, the large hydrometeor cover discrepancy between the KAZR (46.7%) and CloudSat-CPR (27.4%) suggest that the CloudSat-CPR fails to detect clouds in more than a few (on the order of ~40% ) of the atmospheric columns it samples (Fig. 3a). On the other hand, the CloudSat-CPR seems to capture the shape and magnitude of the hydrometeor fraction profile above 1.0 km reasonably well (Fig. 3b). This suggests that the CloudSat-CPR is able to detect the bulk of the thick hydrometeor layers controlling hydrometeor fraction above 1.0 km. This also leads us to believe that the CloudSat-CPR's hydrometeor cover biases results either from its inability to detect clouds entirely located below 1.0 km and/or due to its inability to detect thin and narrow hydrometeor layers that are negligible contributors to hydrometeor fraction. Detailed analysis of the location of individual cloud tops show evidence supporting both of these postulations (Fig. 4a). Specifically: 1) The distribution of KAZR-detected cloud top heights shows clouds below 0.6 which are undetected by the CloudSat-CPR. We estimate that this near-surface cloud mode produces 7.5% of the total cloud cover and so its misdetection could explain nearly half of the CloudSat-CPR hydrometeor cover bias. 2) The distribution of KAZR-detected cloud top heights also shows the presence of cloud top modes near 1.1 and 2.1 km that are only partially detected by the CloudSat-CPR (Fig. 4a).





These elevated cloud tops modes are likely related to the several echo bases between 1.4 and 2.5 km that nearly all
went undetected by the CloudSat-CPR (Fig. 4b). A figure showing time-height observations from two additional
overpass days allows us to visualize that these layers are generally thin, weakly reflective, and broken (Fig. 4i and ii).
We speculate that misdetection of such thin/tenuous clouds explains the remaining of the CloudSat-CPR's cloud cover
bias.
Beyond its inability to detect all cloudy columns, the CloudSat-CPR also severely underestimates the presence of
hydrometeors below 0.75 km because it suffers from surface echo contamination; this creates an artificial enhancement
in the number of apparent hydrometeor layer bases estimated from the CloudSat-CPR near 0.75 km and is not
representative of the true height of the base of either clouds or virga (Fig. 4b). We believe that the surface echo limits
the CloudSat-CPR's ability to observed true cloud base in approximately 52% of the cloudy columns it detects and
true virga base in ~80%; in other words, the CloudSat-CPR often provides an incomplete view of even the WMBL
cloud systems it does detect. This approximation is made based on the subset of cloudy columns observed by the
KAZR whose top is above the CloudSat-CPR surface clutter echo (1.0 km), and that are likely of sufficient thickness
(250 m) and reflectivity (Z > -28 dBZ) to be detected by the CloudSat-CPR.
**4    Challenges**
Although these 89 CloudSat overpasses are reasonably representative of the properties of the WMBL hydrometeor
systems found in the vicinity of the eastern north Atlantic facility, considering the entire set of measurements collected
by KAZR between October 2015 and November 2017 (719 days) provides additional insight on the challenges
associated with measuring the properties of these hydrometeor systems (Fig. 5).
Analysis of the ground-based observations suggests that WMBL cloud fraction exceeds 5% at all heights between 320
m and 2.09 km with cloud fraction peaking at 1.13 km (Fig. 5a; solid black curve). On the other hand, rain tends to be
found in the sub cloud layer below 1.28 km altitude occupying the largest fractional area between 100 m and 1.1 km
(Fig. 5a; dotted black curve). The low height at which WMBL clouds and precipitation are found is especially
challenging for spaceborne system which are known to suffer from contamination from the surface return. We estimate
that roughly 20% of the cloud echoes and 52% of the rain echoes recorded by the KAZR fall within the CloudSat-
CPR's surface echo region which extends at best only to 0.75 km (Fig. 5a; red curves).
The intensity (in terms of radar reflectivity) of cloud and precipitation also largely affects their ability to be detected
by radars.  Using KAZR observations, we characterized the intensity of the hydrometeor echoes observed at each
height and report in Fig. 5b (colormap) the fraction of echoes with a reflectivity above a given threshold at each height.
Generally, cloud and precipitation producing radar reflectivity above a radar MDS can be detected. Thus, we would
expect that the CloudSat-CPR, with its -27dBZ MDS (depicted by the broken black line on Fig. 5b), should have the
capability to detect at best 80% of all cloud and/or echoes forming at any given height, de facto missing at least 20%





of hydrometeor echoes. Radar performance degrades within the surface clutter region. In the clutter region, only those
hydrometeor echoes whose intensity is larger than the surface echo intensity can be detected. To reflect this and for
reference, we overlaid on Fig. 5b the median reflectivity recorded by the CloudSat-CPR in clear sky days between
2010 and 2016 as well as its variability as quantified by the interquartile range (broken and dashed black lines
respectively). Over that time interval, the CloudSat-CPR's median surface echo varied from 37 dBZ at the surface
decreasing to -27 dBZ at 0.75km. Using this curve, we estimate that at 0.5 km height, based simply on sensitivity, the
CloudSat-CPR would miss at least 80% of the echoes detected by KAZR because their reflectivity is below that of the
surface clutter.

Adding to the challenge is the fact that boundary layer systems are shallow. Based on KAZR observations, 53% of
WMBL systems (cloud and rain) forming at ENA are shallower than 500 m, 33% shallower than 250 m and 16%
shallower than 100 m (Fig. 5c; red line). Sampling hydrometeor layers using radar pulses longer than the hydrometeor
layer thickness inherently produces partial beam filling issues, which lead to a weakening of the returned power. This
results in an underestimation of the reflectivity of the thin echoes sampled and may even lead to their misdetection if
the resulting reflectivity is below the radar MDS. There is also an unfortunate relationship between hydrometeor layer
thickness and mean reflectivity such that thin layers not only suffer from more partial beam filling, but also have
weaker reflectivities. The black curve on Fig. 5c shows the median hydrometeor layer mean reflectivity as a function
of hydrometeor layer thickness. From this figure we can estimate that 500 m layer thick hydrometeor layers typically
have a mean reflectivity of -21 dBZ, 250m thick layers -26 dBZ, 100m thick layers -33 dBZ.

## 339 5    Path forward


Improving our ability to detect boundary layer clouds and precipitation could likely be achieved through the following
radar system modifications including (not necessarily in order of importance):

1) Alter the range weighting function
2) Decrease the minimum detectable signal (MDS)
3) Reduce the horizontal field of view
4) Increase the vertical sampling
5) Reduce the transmitted pulse length.
We emulate the impact of these radar modifications by constructing forward-simulations for 7 radar configurations,
each of which has been gradually improved by the aforementioned radar modification (described in Sect. 2.3, Table 1
and Fig. 2). Quantitative assessment of the performance of the forward-simulated radar configurations is estimated
based on a set of 719 forward-simulations constructed from KAZR observations collected between October 2015 and
November 2017. Like done for the real CloudSat-CPR observations in Sect. 3.0, performance is evaluated in terms of
how well hydrometeor cover and hydrometeor fraction are captured (Fig. 7) as well as how accurately the boundaries
of hydrometeor layers are detected (Fig. 8). However, since all forward simulations presented in this section are based
on the same KAZR observations, we expect a perfect match and interpret any deviations from the KAZR observations
as a bias. To help visualize the performance of the 7 radar configurations, we present output from forward-simulations
of the February 27, 2016 hydrometeor layer. The KAZR's view of this hydrometeor layer was depicted and described
in Fig. 1a and Sect. 3.0; for reference the KAZR's detected echo top and base are overlaid on each forward-simulation
in Fig. 6 using black dots.

First, we validate our forward simulation framework by simulating the CloudSat-CPR's current configuration (results
depicted in royal blue and designated as CloudSat$_f$ for short). CloudSat$_f$'s forward simulations show similar biases
than the real CloudSat-CPR when compared to KAZR indicating that the forward simulator captures enough of the
radars characteristics to reasonably emulate its performance. In a nutshell, the CloudSat$_f$ underestimates hydrometeor
cover by more than 10% (Fig. 7a) likely owing to its misdetection of an important fraction of clouds with tops between
750 m and 1.75 km (Fig. 8a) and its inability to detect the small fraction of clouds forming entirely below 500 m. Just
like the real CloudSat-CPR, the CloudSat$_f$ performs well in capturing hydrometeor fraction between 750 m and 3 km
but poorly below that height since it suffers from contamination by surface clutter (Fig. 7b).

Prelaunch testing of the EarthCARE-CPR showed that its pulse generates an asymmetrical point target response. This
mean that, unlike the CloudSat-CPR, the EarthCARE-CPR has an asymmetrical range weighting function (Fig. 2).
The range weighting function of the EarthCARE-CPR's pulse has a rapid cut off at a factor of 0.5 time the pulse length
at its leading edge, and a longer taper extending off to 1.5 times the pulse at its trailing edge. To isolate performance
changes resulting strictly from this range weighting function, we contrast the result of *forward* simulations performed
with the CloudSat-CPR's original configuration (CloudSat$_f$ results depicted in royal blue) and with a CloudSat-like
configuration with the EarthCARE-CPR's *asymmetrical* range weighting function (CloudSat$_a$, results depicted in
cyan). Time-series comparison of CloudSat$_a$ (Fig. 6b) and CloudSat$_f$ (Fig. 6a) reflectivity shows that the asymmetrical
range weighting function reduces the vertical extent of the surface clutter echo, allowing for the detection of a larger
fraction of hydrometeor at 500 m. Over the entire set of 719 forward simulations, this leads to improvements in the
representation of the hydrometeor fraction profile (Fig. 7b) and of the echo base height distribution (not shown) around
500 m. However, differences in the echo base height from KAZR (black dots) and from CloudSat$_a$ (cyan dots) suggest
that changes in the shape of the pulse point target response alone are insufficient to accurately detect the base of the
precipitating WMBL systems found at the ENA (Fig. 6b). We also note that the change in range weighting function
shape alone only marginally improve CloudSat$_f$'s ability to determine hydrometeor cover (improvement from 27.9%
to 28.2% compared to 39.1% reported by KAZR); The reason for this is that hydrometeor cover is controlled by thin,
tenuous clouds and clouds located entirely below 0.5 km. As a potential drawback, the asymmetrical range weighting
function seems to lead to slightly more vertical stretching of cloud top signals (on average 37 m) such as visible by
comparing the examples in Fig. 6a and 6b, and in Fig. 8a. When compounded over the entire ensemble of forward
simulated clouds this leads to a 0.24% overestimation of hydrometeor fraction at all height between 0.75 and 3.00 km
(Fig. 7b). The vertical stretching of cloud tops results from the rapid taper of the pulse between a factor of -0.5-0.0 of



the pulse lengths which is accompanied by additional power being focused in that region of the pulse in contrast to a
symmetrical pulse such as that of the CloudSat-CPR (see Fig. 2).

Besides having an asymmetrical range weighting function, the EarthCARE-CPR will also operate with a MDS of -35
dBZ which is 7 dB more sensitive than the CloudSat-CPR. To isolate performance changes resulting strictly from this
sensitivity enhancement, we contrast the result of forward simulations performed with a CloudSat-like configuration
with the *asymmetrical* range weighting functions (CloudSat$_a$, results depicted in cyan) with that of a CloudSat-like
configuration with both an *asymmetrical* range weighting function *and enhanced sensitivity* (CloudSat$_{a+es}$, results
depicted in purple). Time-series comparison of CloudSat$_{a+es}$ (Fig. 6d) and CloudSat$_a$ (Fig. 6b) reflectivity shows that
the sensitivity enhancement allows for the detection of hydrometeors in previously undetected columns such as the
broken hydrometeor segments observed by KAZR around 100 km distance along the forward-simulated track.
Quantitatively, the more sensitive CloudSat-CPR configuration detects 8% more cloudy columns than either of the
other two CloudSat-CPR configurations discussed so far (i.e., with or without the asymmetrical range weighting
function) missing only 2.4% of the cloudy columns detected by KAZR (Fig. 7a). This implies that, if an important
mission objective is detecting even tenuous cloudy columns, improving the MDS is crucial. That being said, we advise
against accomplishing this by transmitting a longer pulse (e.g., like done in the first 4 years of operation of the GPM-
CPR) since there are two main drawbacks to transmitting a long pulse with a higher sensitivity, both caused by partial
beam filling. Firstly, the enhanced sensitivity leads to additional vertical stretching of cloud boundaries, an effect
visible between 400 and 800 km along track when comparing Fig. 6d to 6b. This is because the signal from cloud
boundaries away from their location resulting from their interaction with the edges of the radar range weighing
function now exceeds the MDS. Secondly, the enhanced sensitivity also leads to previously undetected thin layers
becoming detectable, but it stretches them vertically at least to the vertical extent of the radar pulse length. From
changes in the location of the cloud top height distribution peak shown in Fig. 8a, we estimate that enhancing the
sensitivity of a 3.3 microsecond long pulse from -28 dBZ to -35dBZ would lead to a 250 m bias in detected cloud top
height for the types WMBL clouds forming at the ENA. Moreover, because it both vertically stretches clouds and
detects more real clouds, the highly sensitive CloudSat$_{a+es}$ overestimates hydrometeor cover by up to 7% at all heights
between 500 m and 3.0 km (Fig. 7b).

Since EarthCARE will travel at an altitude closer to the Earth surface it will also have half the horizontal field of view
of CloudSat. Our results suggest that halving the CloudSat-CPR's horizontal field of view and halving its integration
distance would lead to a slight reduction in its estimated hydrometeor cover (1.7% less). We take this as an indication
that the larger horizontal field of view of the CloudSat-CPR only marginally artificially broadens broken clouds (see
CloudSat$_{a+es+hf}$, results depicted in gold in Fig. 7). That being said, note that this result, like all the others presented
here, is based on 2-D forward-simulation and as such it does not take into account cross-track effects which may also
generate biases especially in sparse broken cloud fields.

Another interesting radar configuration proposed by the EarthCARE mission advisory group concerns the amount of



vertical oversampling of the radar pulse. Radar signals are typically oversampled by a factor of two effectively halving
the vertical spacing between available measurements. The EarthCARE-CPR will use a factor of 5 oversampling to
increase its vertical range sampling to 100 m while still operating at a 500 m vertical resolution. While oversampling
may be appealing because it creates a smoother view of cloud fields, it does not effectively improve the vertical
resolution because of the correlations between the oversampled measurements. Evaluating the impact of these
correlations on the observed radar reflectivity field is beyond the scope of this study which instead focuses on
evaluating the impact of oversampling on accurately locating cloud and precipitation boundaries. Time-series of
EarthCARE (Fig. 6b) reflectivity shows that increased oversampling will allows for a more precise characterization
of the variability of echo base and top height (also see the echo top height distribution presented in Fig. 8c).
Comparison of the ensemble of EarthCARE (magenta) and CloudSat$_{a+es+hf}$ (gold) forward-simulations indicates that
this precision can be achieved without causing significant biases in hydrometeor cover (Fig. 7a) or hydrometeor
fraction (Fig. 7c).
Although the EarthCARE-CPR's performance is significantly better than that of the CloudSat-CPR when it comes to
detecting thin, tenuous and broken clouds as well as clouds and precipitation near 500 m, its configuration still does
not allow to detect all WMBL clouds and precipitation. Remaining detection limitations occur below 500 m within
the region of the surface clutter echo. Additional reduction of the vertical extent of the surface clutter can be achieved
by reducing the pulse length. This, however, comes at the expense of reduced sensitivity. Comparing EarthCARE
(results depicted in magenta), ACCP$_{250}$ (results depicted in red) and ACCP$_{100}$ (results depicted in green) simulations
allows us to see the gain and penalty incurred from shortening the radar vertical range resolution from 500 m, to 250
m to 100 m at the cost of reducing sensitivity from -35 dBZ to -26 dBZ and -17dBZ. In alignment with our previous
conclusion that a high sensitivity is necessary for detecting all cloudy columns, reducing the radar pulse length and
sensitivity reduces the fraction of cloudy columns which can be detected by the ACCP configurations (Fig. 7a). For
instance, the ACCP$_{250}$ configuration, which is nearly as sensitive as CloudSat (-26 dB versus -28 dB), performs very
similarly in terms of the number of cloudy columns it is able to detect (Fig. 7a) and in terms of how well it can capture
the vertical distribution of hydrometeors between 500 m and 3.0 km (Fig. 7d) which we determined is influenced by
the deeper more reflective clouds rather than the thin and tenuous ones. The ACCP$_{250}$ configuration does, however,
have the advantage of providing information on the base of clouds and/or precipitation down to 250 m which is much
more than the CloudSat-CPR can achieve (Fig. 7d). ACCP$_{250}$'s shorter pulse also helps mitigate the amount of cloud
stretching related to partial beam filling issues thus providing a more precise characterization of cloud top height (Fig.
8c, effects also visible in Fig. 6e). So generally speaking, reducing vertical pulse length reduces the fraction of detected
cloudy columns but improves the characterization (both in terms of echo top and echo base location) of those cloudy
columns which are detected.
Results also suggest that radars with shorter less sensitive pulses would be more suitable for the characterization of
surface rain and virga, which are more reflective targets. In fact, we estimate that ACCP$_{100}$ would detect 18% out of
the 26% rainy columns detected by the KAZR (Fig. 7a). ACCP$_{100}$ would also do reasonably well at capturing the



vertical distribution of drizzle and rain; comparisons of rain fraction profiles estimated from the KAZR (subcloud
layer only) suggest that $ACCP_{100}$ would miss < 2% of the virga forming at each height below 750 m and would be
able to detect the presence of rain as close as 25 m from the surface.

**6    Discussion and conclusions**

The macrophysical properties of warm marine boundary layer (WMBL) clouds and precipitation and spaceborne
radars ability to characterize them is evaluated using ground-based ceilometer and Ka-band ARM Zenith Radar
(KAZR) observations collected over the Atmospheric Radiation Measurement (ARM) program Eastern North Atlantic
(ENA) facility.

Analysis of 719 days of KAZR observations collected between October 2015 and November 2017 suggest that the
following three main properties of WMBL clouds and precipitation complicate their detection by spaceborne radars:

1)    They are generally thin, with 50 % of the hydrometeors layer detected by KAZR having a thickness below
400 m. As a result, they may not fill the entire spaceborne radar pulse volumes causing serious partial beam
filling issues.

2)    They are weakly reflective, with 50 % of the hydrometeors detected by KAZR having reflectivity below -22
dBZ. We also find that hydrometeor layer mean reflectivity is strongly related to hydrometeor layer thickness
such than the thinnest layers are also typically the least reflective ones, further challenging their detection.

3)    They form at low levels, with 50% of WMBL cloud echoes being located below 1.2 km and 50 % of sub-
cloud layer rain echoes below 0.75 km. Therefore, their backscattered power may easily overlap and be
masked by the strong surface return detected by spaceborne radars.

Observations from 89 daytime overpasses and results from 719 2-D forward simulations constructed using KAZR
observations consistently shows that the CloudSat-CPR fails to detect 29-41% of the cloudy columns detected by the
ground based KAZR. Supporting the postulations of both *Christensen et al.* [2013], *Rapp et al.* [2013] and *Liu et al.*
[2018],  our results suggest that a little over half of this bias can be attributed to the CloudSat-CPR inability to sample
thin, tenuous cloud while the other half results from misdetection of clouds that form entirely within the CloudSat-
CPR surface (some of which are also thin and tenuous). Using forward simulations, we determined that mitigating the
vertical extent of the surface clutter by changing its range weighing function or by reducing its vertical range resolution
by half would only partially improve the CloudSat-CPR's ability to detect all cloudy columns, which is very much
limited by the CloudSat-CPR's low sensitivity. In other words, when it comes to detecting all cloudy columns, we
find that improving radar MDS is more important than reducing the vertical extent of the surface clutter. For this
reason, the 7 dB more sensitive EarthCARE-CPR is expected to detect significantly (19.7%) more cloudy columns
than the CloudSat-CPR, only missing < 9.0% of the simulated cloudy columns.



On the other hand, our overpass and forward-simulation results also suggest that the CloudSat-CPR is able to capture
the general vertical distribution of hydrometeor (i.e., hydrometeor fraction profile) above 750 m which we find is
dominantly controlled by thicker more reflective clouds. Unfortunately, we estimate that because of its asymmetrical
range weighting function and because of the long length of his highly sensitive pulse, the EarthCARE-CPR's will
overestimate (by ~250 m) cloud top height and underestimate cloud base height, making hydrometeor layers appear
artificially thicker than they are, which will also bias the EarthCARE-CPR's hydrometeor fraction estimates. This
effect would need to be addressed to extract accurate information about the location of cloud boundaries and about
the vertical distribution of clouds and precipitation, two aspects likely to become increasingly important as we continue
moving towards increasingly high-resolution global modeling. Synergy with a collocated ceilometer could potentially
help correct cloud top height, however, such corrections would only be possible in single layer conditions and
alternative techniques would need to be developed to improve the EarthCARE-CPR's ability to accurately estimate
the vertical extent of multi-layer boundary layer clouds.

Below 1.0 km, the surface clutter echo seen by the CloudSat-CPR masks portions of clouds and virga. Based on a
subset of KAZR observations, we estimate that the surface echo limits the CloudSat-CPR's ability to observed true
cloud base in ~52% of the cloudy columns it detects and true virga base in ~80%. In other words, the CloudSat-CPR
often provides an incomplete view of even these cloud systems it does detect. Our analysis of real CloudSat-CPR's
observations shows that the clutter mask part of the GEOPROF version 4.0 product is relatively aggressive, and we
believe the CloudSat-CPR's performance could perhaps be somewhat improved by revising this clutter mask. In terms
of future spaceborne radar missions, radar architectures with finer range resolution could more precisely characterize
the boundaries of hydrometeor layers. For instance, the 250-m range resolution (oversampled at 125-m) radar
architecture presented here produces echo top height statistics comparable to that of the ground based KAZR in terms
of detecting the minimum, maximum and mode of the distributions. However, since a shorter pulse can currently only
be achieved at the expense of reduced sensitivity, this radar would suffer from the limitations similar to that of the
CloudSat-CPR in terms of the number of cloudy columns it could detect. This means that while improving the
detection of virga below 500 m might be possible, improving the detection of cloud bases below 500 m is unlikely
achievable with current technologies.

Overall this analysis suggests that no one single radar configuration can adequately detect all WMBL clouds while
simultaneously accurately determining the height of cloud top, cloud base and virga base. The alternative of deploying
spaceborne radars capable of operating with interlaced operation modes is thus worth considering [*Kollias et al.*,
2007]. For example, a radar capable of generating both a highly sensitive long-pulse mode and a less sensitive but
clutter limiting short-pulse mode would likely provide a more comprehensive characterization of the boundary layer
by detecting both low-reflectivity clouds and low-altitude rain.

On a related note, it is likely that the partial beam filling issues identified here as affecting both the CloudSat-CPR
and the EarthCARE-CPR ability to locate clouds might, as hinted by *Burns et al.* [2016], also affect their ability to



accurately measure their true reflectivity. Such radar reflectivity biases would affect water mass retrievals performed
using radar reflectivity measurement and follow up efforts should aim at quantifying this effect and should look into
alternative retrieval techniques and/or radar configurations that could address this issue [*Battaglia et al.*, In
preparation].

As a final thought we also point out that, due to the variations in the microphysical and macrophysical properties of
oceanic warm clouds globally, the actual missed detections by the various spaceborne-CPR architectures described
here may change when considering other regimes. *Liu et al.* [2016] study hints at the fact that regions dominated by
stratiform clouds are more challenging to characterize than those dominated by cumulus. Thus, for completeness,
follow on studies could test the performance of the radar configurations proposed here in other climatic regimes.

**Authors contributions**

K. Lamer coordinated the project, extracted the ground-based measurement files from the ARM archive, performed
the data analysis and produced the final manuscript draft. P. Kollias extracted the CloudSat-CPR GEOPROF product
files from the data processing center and provided feedback on the forward-simulator. A. Battaglia provided feedback
on the analysis methods as well as on the manuscript draft. S. Preval performed exploratory data analysis and provided
feedback on the manuscript draft.

**Acknowledgements**

K. Lamer's contributions were supported by U.S. Department of Energy Atmospheric System Research project DE-
SC0016344. P. Kollias's contributions were supported by the U.S. Department of Energy Atmospheric Systems
Research program and the ENA site scientist award. A. Battaglia and S. Preval's contributions were supported by the
U.S. Department of Energy Atmospheric System Research project DE-SC0017967.

**Data Availability**

All CloudSat-CPR observations were obtained from the CloudSat data processing center (www.
http://www.cloudsat.cira.colostate.edu/). All ARM observations were obtained from the ARM archive
(https://www.archive.arm.gov/discovery/). Output of all forward-simulations is fully reproducible from the
information given.








**Appendix I**
Since the Earth surface can be treated as a point target, observations of the surface clutter echo during clear sky
conditions can be used to gain insight into how the energy contained within radar pulse spreads out vertically when it
hits a point target (i.e. about range-weighting function).
We extract information about the shape of the CloudSat-CPR's range-weighting function from a subset of observations
collected between May 2010 and November 2017 identified as clear sky in the GEOPROF product (version 4.0;
CPR_Echo_Top mask variable). We further ignore observations from non-significant echoes (Z < -27 dBZ) and
mispointing events (profiles, which have their maximum reflectivity more than 75 m from 0 m height). Over this
period, the median surface reflectivity profile (depicted by the broken black profile in Fig. 5c) shows a main peak at
surface level quickly reducing in intensity within height; the surface radar reflectivity return was observed to reduce
by ~34 dB at a distance of 0.5 km (i.e., half the pulse length) away from it actual location at the surface. A secondary
lobe whose peak intensity is ~50 dB lower than that of the main lobe was observed to spread from a distance of roughly
0.5 km to 1.0 km away from the main peak. Characterization of the CloudSat-CPR point-target response presented in
*Tanelli et al.* [2008] also revealed the symmetrical character of the main lobe of the CloudSat-CPR range-weighting
function; the prelaunch analysis also showed that the presence of this secondary is confined to the pulse's leading
edge.
In the current analysis, we first use the median surface reflectivity profile we extracted (post-launch) to adjust the
width of the gaussian range weighting function used in the CloudSat forward-simulator. The gaussian range weighting
function depicted in Fig. 2 produces a forward-simulated surface echo return similar, in intensity and vertical extent,
to the surface echo observed by the CloudSat-CPR under clear sky conditions (compare the royal blue line and black
lines in Fig. 5b). Note that we did not attempt to reproduce the CloudSat-CPR's secondary lobe and that the use of
this gaussian range weighting function is limited to the CloudSat$_f$ forward simulation. All other forward simulations
are conducted using the EarthCARE-CPR asymmetrical range weighting function constructed from pre-launch testing
of the EarthCARE-CPR.
The strength of the surface echo observed by CloudSat under clear sky conditions is also used to determine the
intensity of the surface clutter artificially input to the KAZR reflectivity field. We estimate the surface echo to be
added to KAZR's -30 m to 0 m range gate should have an intensity of 52 dBZ such that after its convolution by the
range weighting functions of the spaceborne radar configurations, the strength of the realized surface echo at 0 m
height is 41 dBZ matching the strength of the surface echo observed by CloudSat under clear sky conditions (depicted
by the broken black line in Fig. 5b). Note that variability of the surface return due to attenuation of the radar signal by
liquid, heterogeneous surface conditions, and changes in satellite altitude have not been included in the forward-
simulator. However, analysis of the real CloudSat surface echo observed during clear sky suggest that variability due
to heterogeneous surface conditions, and changes in satellite altitude are on the order of <2 dB (depicted by the dotted
black lines in Fig. 5b).



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



**Tables**

**Table 1.** Specifications of the forward-simulated radar configurations including information about whether or not their
pulse weighting function is symmetrical (sym.) or asymmetrical (asym.) in either the vertical or the along-track
dimension.

| Forward-simulated radar architectures | Sensitivity (dBZ) | Vertical dimension | | | | | Along-track dimension | | |
|---|---|---|---|---|---|---|---|---|---|
| | | Pulse length (km) | Range resolution 6-dB (m) | Oversampling | Range sampling (m) | Range weighting function shape | Instantaneous field of view (km) | Integration distance (km) | Weighting function shape |
| CloudSat$_f$ | -28 | 1.0 | 500 | 2 | 250 | Sym.* | 1.4 | 1.0 | Sym. |
| CloudSat$_a$ | -28 | 1.0 | 500 | 2 | 250 | Asym* | 1.4 | 1.0 | Sym. |
| CloudSat$_{a+es}$ | -35 | 1.0 | 500 | 2 | 250 | Asym* | 1.4 | 1.0 | Sym. |
| CloudSat$_{a+es+hhf}$ | -35 | 1.0 | 500 | 2 | 250 | Asym* | 0.7 | 0.5 | Sym. |
| EarthCARE | -35 | 1.0 | 500 | 5 | 100 | Asym* | 0.7 | 0.5 | Sym. |
| ACCP$_{250}$ | -26 | 0.5 | 250 | 2 | 125 | Asym* | 0.7 | 0.5 | Sym. |
| ACCP$_{100}$ | -17 | 0.2 | 100 | 2 | 50 | Asym* | 0.7 | 0.5 | Sym. |

* Shape of the range weighting function is depicted in Fig. 2
** Across track dimension is not represented

**Figures**

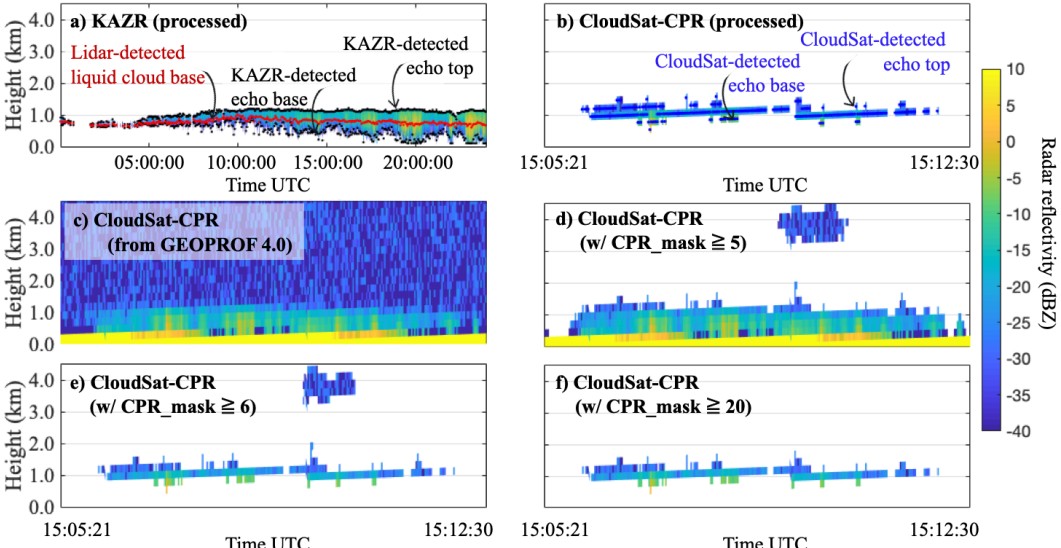


**Figure 1.** Hydrometeor radar reflectivity measured on Feb. 27, 2016 a) by KAZR and b) by the CloudSat-CPR when
it overpassed the KAZR located at the Eastern North Atlantic (ENA) observatory at 15:05:21 UTC. For KAZR, 24-
hrs of measurements are show. For CloudSat, a ground-track taken in ~7-sec is shown (a total length of ~3,000 km).
Dots on these figures represent the boundaries of the radar echo (black and blue dots for the KAZR and the CloudSat-
CPR respectively) and the location of the ceilometer-determined cloud base (red dots). Also plotted are the CloudSat
radar reflectivity c) raw, d) for significant returns (CPR_mask >5), e) for echoes deemed very weak and stronger
(CPR_mask > 6) and f) for echoes deemed weak and stronger (CPR_mask > 20).








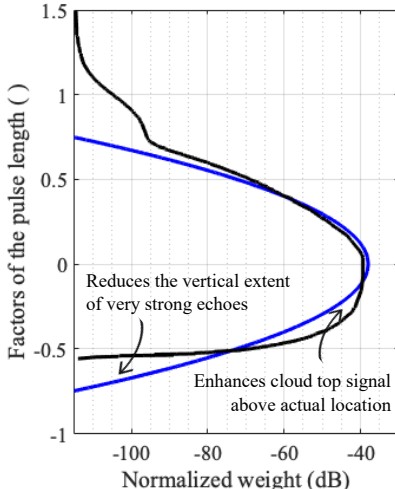

**Figure 2**. Symmetrical (blue) and asymmetrical (black) range weighting functions for the forward simulated radar architectures detailed in Table 1.

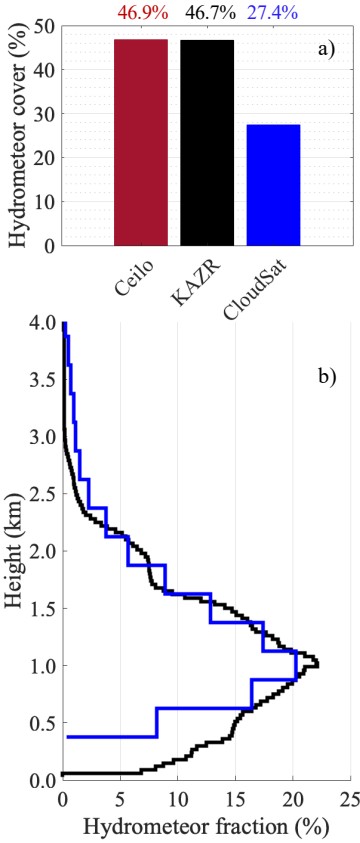

748

**Figure 3.** For 89 days where CloudSat overpassed the ENA observatory, a) fraction of observed profiles with cloud

or rain (i.e., hydrometeor cover) and b) hydrometeor fraction profile. Both estimated from CloudSat-CPR observations

(blue) and ground based KAZR observations during the 4-hr time window when CloudSat overpassed the KAZR

(black).



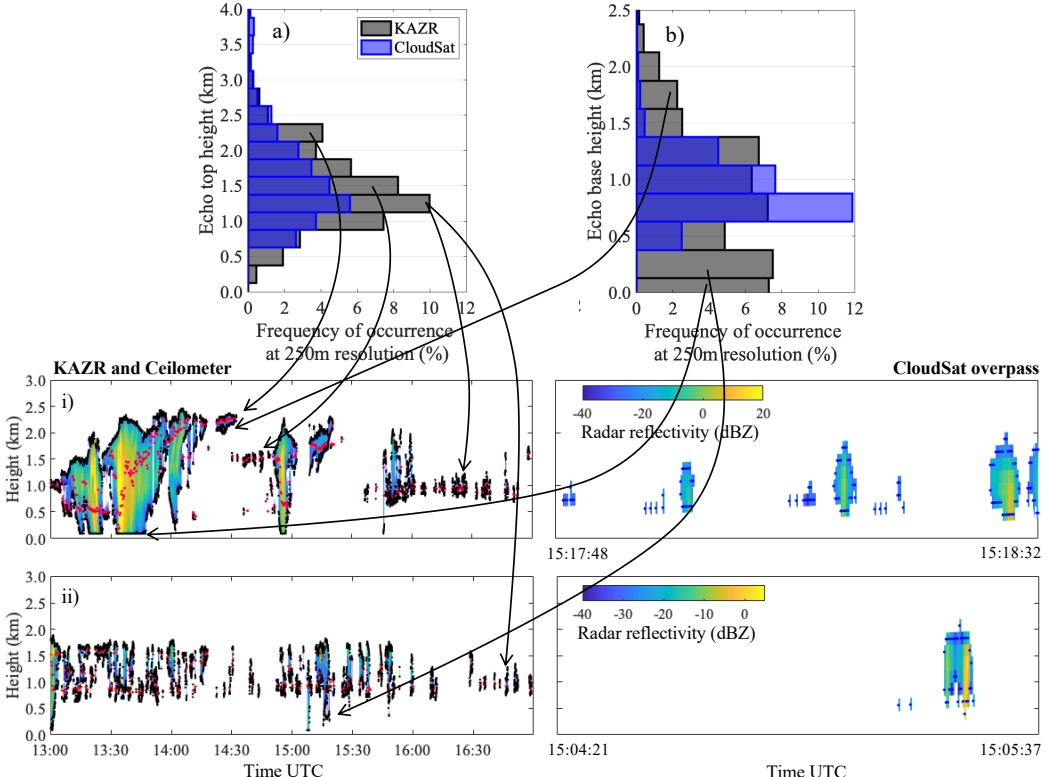

**Figure 4.** For 89 days where CloudSat overpassed in the vicinity of the ENA observatory, distribution of a) echo base height, and b) echo top height, estimated from CloudSat-CPR observations (blue) and ground-based KAZR observation during the 4-hr time window when the CloudSat-CPR overpassed the KAZR (grey). For references are examples of hydrometeor radar reflectivity measured on i) Feb. 02, 2017 and ii) Oct. 24, 2016 by the ground based KAZR and by the CloudSat-CPR. Dots on these figures represent the boundaries of the radar echo (black and blue dots for the KAZR and the CloudSat-CPR respectively) and the location of the ceilometer-determined cloud base (red dots).

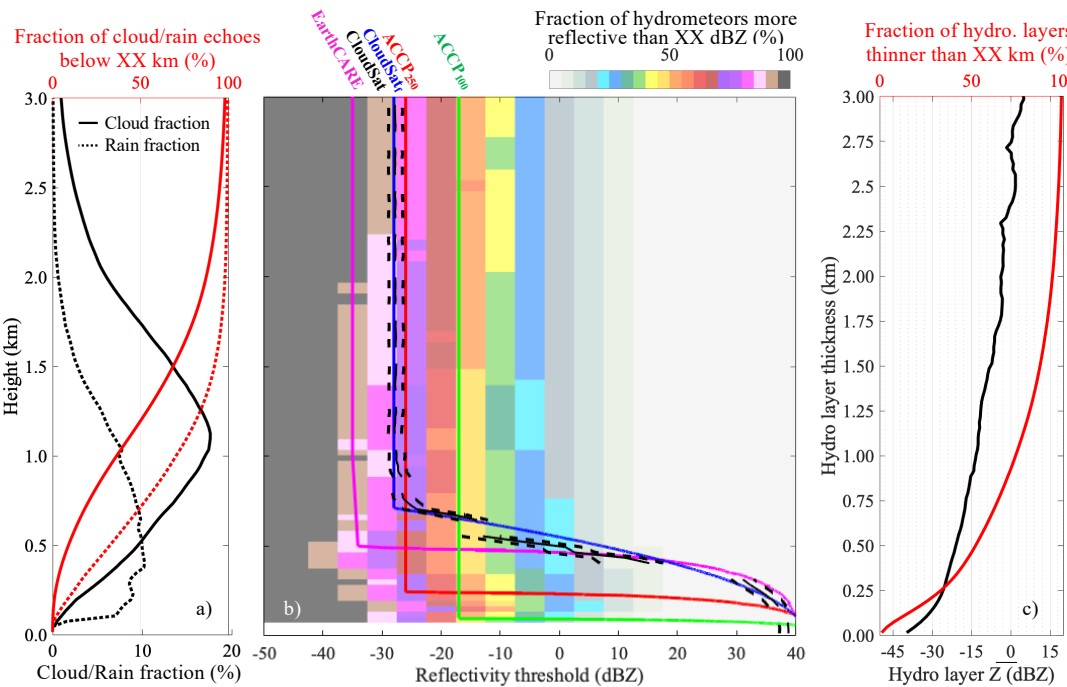

**Figure 5.** From ground based KAZR observations collected between 10/2015 and 02/2018, a) profile of cloud (solid
black line) and sub-cloud layer rain (dotted black line) fraction, and the fraction of either cloud (solid red line) or sub-
cloud-layer rain (dotted red line) echoes located below of certain height. b) Fraction of hydrometeor (cloud or rain)
echoes with reflectivity larger than a given reflectivity threshold (colormap) with superimposed the surface clutter
profile as simulated for the CloudSat (royal blue line) EarthCARE (magenta line), ACCP$_{250}$ (red line) and ACCP$_{100}$
(green line) CPR configurations and as observed by the CloudSat-CPR between May 2010 and November 2017
(broken black line marks the median, dotted black lines mark the interquartile range); c) median profile of hydrometeor
layer mean reflectivity as a function of thickness (black) and the fraction hydrometeor (cloud and rain) layers thinner
than a certain thickness (red).

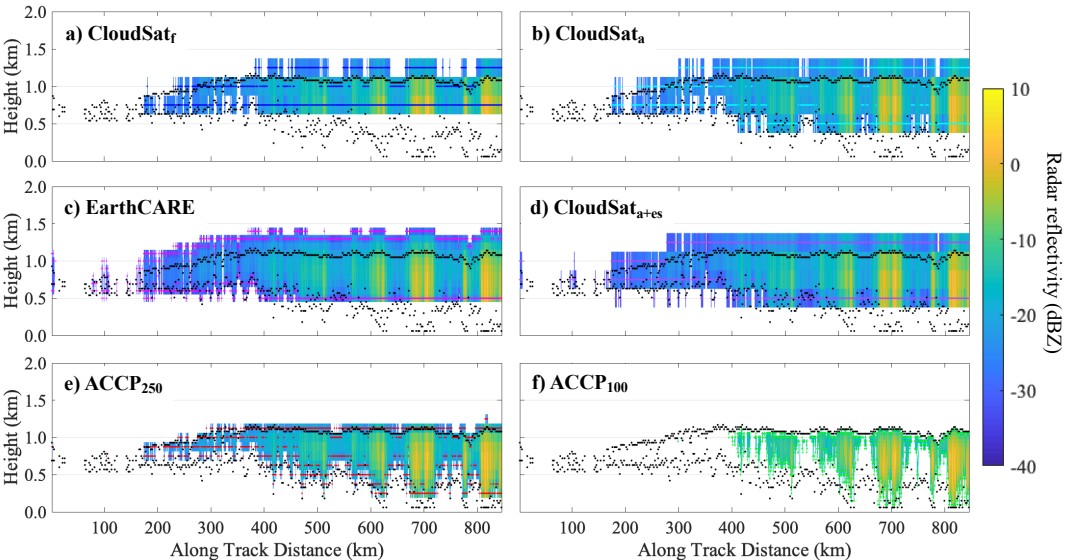

780

**Figure 6.** Based on KAZR observations of the hydrometeor layer of Feb. 27, 2016, forward simulated radar reflectivity

(colormap) and estimated hydrometeor layer boundaries (colored dots) for a) $CloudSat_f$ (royal blue dots), b)

$CloudSat_{nps}$ which is CloudSat operating with the EarthCARE asymmetrical range weighting function (cyan dots), d)

$CloudSat_{nps+es}$ which additionally has an enhanced sensitivity equivalent to the EarthCARE (purple dots), c)

EarthCARE which additionally operates with a factor of 5 vertical oversampling (magenta dots), e) $ACCP_{250}$ which

instead has a 250-m range resolution (red dots) and f) $ACCP_{100}$ which instead has a 100-m range resolution (green

dots). For reference, the corresponding KAZR observed radar reflectivity are depicted in Fig. 1a and echo boundaries

identified by the KAZR are overlaid on each subpanel using black dots.

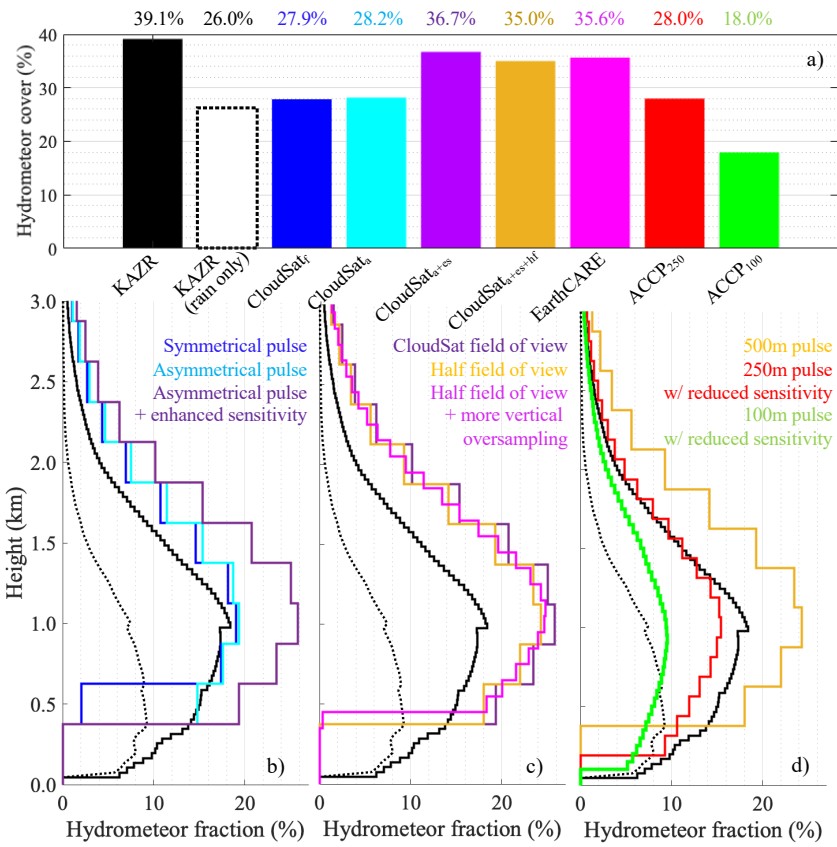

**Figure 7.** For 719 forward simulated days: a) fraction of observed profiles containing either cloud or rain (i.e., hydrometeor cover); Also, for KAZR only, using complementary ceilometer observations, we estimate the fraction of all observed profiles containing rain in the sub-cloud layer. b-c-d) hydrometeor fraction profile estimated for all the forward-simulated radar architectures. All acronyms and colors are defined in Fig. 6 with the exception of CloudSat$_{nps+es+hf}$ which is a the CloudSat operating with EarthCARE's asymmetrical range weighting function, enhanced sensitivity and half the horizontal field of view (gold).



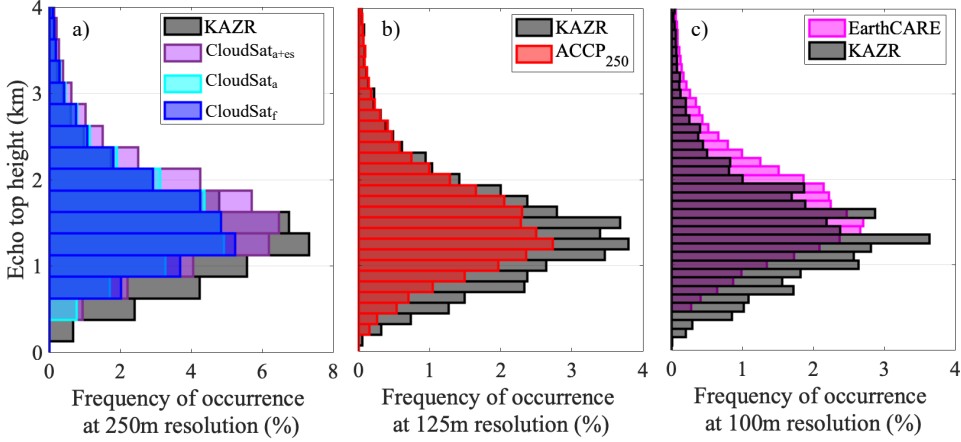

**Figure 8.** For 719 forward simulated days, distribution of echo top height observed by KAZR (grey) and estimated from the forward simulated radar architectures. Results are estimated at various range sampling resolutions according to the capability each spaceborne sensor configuration. All acronyms and colors are defined in Fig. 6.