# Peer review of "Mind-the-gap part I: Accurately locating warm marine boundary layer clouds and precipitation using spaceborne radars"

_Atmospheric Measurement Techniques, 2019_

## Referee Comment (RC1) · Anonymous Referee #1 · 10 Feb 2020

This study makes use of a long deployment of extremely sensitive ground-based ARM radar to provide new and important insights into the factors limiting the detection of warm marine boundary layer clouds by CloudSat. By considering the independent effects of the radar pulse response function, sensitivity, field of view, vertical over-sampling, and pulse length, the authors provide important comparisons between the performance to be expected from future EarthCARE and NASA ACCP cloud profiling radars. This work both improves our understanding of existing and widely used Cloud-Sat datasets, and will help in the preparation for future cloud radar missions.

The manuscript is logically structured and clearly written. Subject to a few corrections

for clarity, I recommend this manuscript for publication in Atmospheric Measurement Techniques.

Specific comments:

P1, L10: not sure if there is a word missing, or just that the word after the semi-colon shouldn't be capitalised.

P1, L16: it's clear to me what you mean, but "shortening" surface clutter is an awkward phrase.

P4 L101–103: this sentence is hard to parse, and could do with restructuring or perhaps replacing the em-dashes with commas. Something like, "... information from CloudSat-CPR to evaluation the performance of current spaceborne sensors in this regime (Section 2.1), ARM measurements used as a benchmark (Section 2.2), and how we forward-simulate..."

P4, L107: referring to CloudSat making observations "twice a day" or "once a day" is misleading; this refers to the day-time and night-time parts of a CloudSat orbit, of which there are many each day.

P5, L160–164: is it possible to use KAZR measurements to comment on how conservative (or aggressive) this approach to clutter filtering is? The argument is made in the conclusions (P15, L518—520) that improvements are possible, but I couldn't find (and this may easily be my oversight, and if so I apologise) where this was stated in the results section.

P7 , L212: the word after a semi-colon shouldn't be capitalised.

P8, L249: When discussing times it is clearest to state that all times are in UTC, but also provide important information about local time so we know what to expect with respect to the diurnal cycle.

P8, L258: remove "both"

[Figure]

P10, L317: If I understand the caption of Fig. 5 correctly, CloudSat-CPR is shown by a royal blue line.

P11, L373: should read "...a factor of 0.5 times the pulse length..."

P13, L436: should be Fig. 6c

P14, L474–3: should be something like "...warm marine boundary layer (WMBL) clouds and precipitation, and spaceborne radars' ability to characterisze them, is..."

P14, L485: should be "...such that..."

P14, L491: remove "both"

P15, L505: "...length of its hightly sensitive pulse..."

P16, L546: remove "study"

P17, L583: should be Fig. 5b

P17, L589, should be "...this secondary lobe is confined..."

P22, L728, Figure 1: should be "...ground-track taken in $\sim$7 minutes is shown..."

Figure 1: I (a color-blind reader) have a lot of difficulty distinguishing the blue dots in Figure 5b from the underlying radar reflectivity (also blue). Since they are not on the same subplot, would it be acceptable to make these dots black as well?

Figure 1: To make clear the fact that the KAZR and CPR data are on different time-series, it may be useful to mark the time of the CPR overpass with a vertical line on the KAZR timeseries. This would also aid comparison of the cloud fields at the same time.

Figure 2: The y-label "factors of the pulse length" is unclear; the label and the sign convention should make it very clear which is the "leading edge" and the which the "trailing edge" of the radar pulse in the direction of propogation.

P24, L745–748, Figure 3: In the text is seemed clear that these values (e.g. hydrometeor cover) are fractions of profiles excluding those containing high, mid- and layered clouds. If so, best to re-state this in the caption.

Figure 4: It may again be useful to show the time of the CloudSat overpass on the KAZR timeseries.

P26, L 766, Figure 5: should be "...located below a certain height."

P28, L795, Figure 7: "...which is CloudSat operating with..."

---

## Referee Comment (RC2) · Anonymous Referee #2 · 12 Feb 2020

The paper (a) discusses the factors limiting CloudSat-CPR to detect warm marine boundary layer (WMBL) clouds, (b) quantifies the CloudSat ability to accurately estimate their characteristics (coverage, vertical distribution, top/bottom boundaries) using long-term ground-based measurements from the KAZAR–ARM radar, and (c) evaluates the performance of 7 alternative configurations for CloudSat, EarthCARE and ACCP CPR observations for an optimum characterization of these clouds (specifically the cloud reflectivity and the hydrometeor boundaries) by comparing forward simulations from the different configurations with the KAZAR forward simulations. This work improves our understanding on (a) the performance of CloudSat dataset on WBML clouds, (b) the performance of the future cloud radars onboard EarthCARE and ACCP

and (c) the differences of the performances of the 3 instruments. Additionally it provides recommendation for the next generation of space-borne radars targeting WMBL science. This is a very interesting aspect and the paper is well structured and clearly written, for that reason I believe that this paper is appropriate for publication in Atmospheric Measurement Techniques.

I have only one comment, in the statistics of hydrometeor layer properties estimated for days where CloudSat overpassed within 200 km of the ENA station, 4 hrs KAZR and ceilometer observations around the overpass are taken into consideration (Figures 3 and 4). Why do the authors use such a wide time window for their comparison when for cloud-comparison purposes, a length scale of a few tens of kilometers and a time scale of a few minutes is generally acceptable (e.g. Blanchard et al., 2014)? This question is more puzzling in the discussion of the limitations of CloudSat observations, highlighted in Figure 4, with cloud observations up to 1:30 hour time difference with the time of the overpass. I suggest that the authors use a smaller time window for the evaluation of CloudSat performance with KAZAR measurements and provide a justification for the use of this time window and the consequences on the homogeneity of the scene. Similarly, in the discussion of the differences of the statistics observed, it would be good if the examples/arrows pointing to the different CloudSat underestimations/limitations are given in cases that these limitations are visible in the clouds captured from CloudSat and KAZAR collocated cloud observations.

The rest of my specific comments are only to encourage more clarity in the presentation of the results or technical corrections.

1. Page 4, line 313 – 326: Although mentioned in the legend of Figure 5b, the CloudSat blue line in fig. 5b is not mention in the paragraph.

2. Page 11, line 391: There is a typo in the factor.

3. Page 15, line 510: Apart from a ceilometer, the synergy with the EarthCARE lidar (ATLID) could help correct the cloud top height.

Reference: Blanchard, Y., J. Pelon, E.W. Eloranta, K.P. Moran, J. Delanoë, and G. Sèze, 2014: A Synergistic Analysis of Cloud Cover and Vertical Distribution from A-Train and Ground-Based Sensors over the High Arctic Station Eureka from 2006 to 2010. J. Appl. Meteor. Climatol., 53, 2553–2570, https://doi.org/10.1175/JAMC-D-14-0021.1.

---

## Referee Comment (RC3) · Anonymous Referee #3 · 13 Feb 2020

This study clearly demonstrates the capabilities and constraints regarding spaceborne detection of the warm marine boundary layer (WMBL) clouds. This is crucial information in terms of the expected uncertainties in weather and climate projections and especially on the role of WMBL layers in radiation budget. The work is clearly written and the comparison between KAZAR–ARM ground based and CloudSat radars provides significant insight on the expected products from EarthCARE and ACCP missions as well as possible recommendations for future missions such as the combination of both short and long pulse modes for detecting WMBL clouds along with their macrophysical properties. Therefore I recommend publication of this work in Atmospheric Measurement Techniques.

---

## Author Comment (AC1) · 3 Apr 2020

The authors would like to thank all reviewers for their kind words and feedback. A point-by-point response to the reviewer's comments is provided below.

**Reviewer 1**

**Specific comments:**

**P1, L10: not sure if there is a word missing, or just that the word after the semi-colon shouldn't be capitalized.**

All instances of capitals following semi-colons have been corrected.

**P1, L16: it's clear to me what you mean, but "shortening" surface clutter is an awkward phrase.**

Following the reviewer's suggestion, we have revised this senstence:

*"Out of all configurations tested, the 7 dB more sensitive EarthCARE-CPR performs best (only missing 9.0% of cloudy columns) indicating that improving radar sensitivity is more important than* decreasing the vertical extent of *surface clutter for observing cloud cover."*

**P4 L101–103: this sentence is hard to parse and could do with restructuring or perhaps replacing the em-dashes with commas. Something like, "... information from CloudSat-CPR to evaluation the performance of current spaceborne sensors in this regime (Section 2.1), ARM measurements used as a benchmark (Section 2.2), and how we forward-simulate..."**

Following the reviewer's suggestion, we have revised this paragraph:

*"The next sub-sections describe how we extracted cloud and precipitation information from raw CloudSat-CPR to evaluate its performance (Sect. 2.1), ARM measurements which act as a benchmark (Sect. 2.2) and how we forward-simulated alternative spaceborne radar configurations (Sect. 2.3)."*

**P4, L107: referring to CloudSat making observations "twice a day" or "once a day" is misleading; this refers to the day-time and night-time parts of a CloudSat orbit, of which there are many each day.**

Following the reviewer's suggestion, we have revised this paragraph:

*"The CloudSat-CPR has been collecting observations since May 2006. It follows a sun-synchronous orbit set to cross the equator at 13:30 local mean time, repeating its ground track every 16 days. The CloudSat-CPR went offline between May and October 2011 because of a spacecraft battery failure. After it returned online, it was placed in daylight-only mode [Stephens et al., 2018]. Periods when CloudSat passed within a 200 km radius of the ARM ENA ground-based facility are used to evaluate the CloudSat-CPR's ability to characterize WMBL clouds and precipitation (results presented in Sect. 3.0); this happened on 138 instances since the ground-*

*based site was made permanent at the end of 2015. For this site, daylight-mode operations make it such that data is collected only around 15:00 UTC between August and April but at both 4:00 and 15:00 UTC between May and July."*

**P5, L160–164: is it possible to use KAZR measurements to comment on how conservative (or aggressive) this approach to clutter filtering is? The argument is made in the conclusions (P15, L518) that improvements are possible, but I couldn't find (and this may easily be my oversight, and if so I apologies) where this was stated in the results section.**

To clarify, the authors made this argument based on their visual comparison of raw and masked CloudSat-CPR observations and not because of comparison to KAZR. The text was revised accordingly:

*"Comparison of raw and masked CloudSat-CPR's observations suggest that the clutter mask part of the GEOPROF version 4.0 product is relatively aggressive, and we believe the CloudSat-CPR's performance could perhaps be somewhat improved by revising this clutter mask; That being said a sensitivity study of the thresholds in the CloudSat-CPR clutter mask is beyond the scope of this study."*

A one-to-one comparison between the CloudSat-CPR and a "truth" would ideally be used to quantify the performance of the clutter mask. Since the KAZR and the CloudSat-CPR do not have the same temporal and spatial resolution, the KAZR cannot directly be used as "truth"; as such this effort would require the development of appropriate methods beyond the scope of this study.

**P7, L212: the word after a semi-colon shouldn't be capitalized.**

All instances of capitals following semi-colons have been corrected.

**P8, L249: When discussing times it is clearest to state that all times are in UTC, but also provide important information about local time so we know what to expect with respect to the diurnal cycle.**

All mentions of time are now accompanied by "UTC". Graciosa's local time is "UTC-1" during the winter months and "match UTC" during the daylight savings time months. This means that for the particular example presented in Fig. 1 local time is "UTC-1". The first sentence of the paragraph referring to this figure was modified to provide this information:

*"Between 0:00 and 10:00 UTC (23:00 and 9:00 local time), cloud top height was observed to rise at a rate of roughly 21m hr$^{-1}$."*

**P8, L258: remove "both"**

The word "both" was removed.

**P10, L317: If I understand the caption of Fig. 5 correctly, CloudSat-CPR is shown by a royal blue line.**

The royal blue line on Fig. 5b represents: *"the surface clutter profile as simulated for the CloudSat (royal blue line)"*. While the broken black line and dotted black lines on Fig. 5b represents: the surface clutter profile *"as observed by the CloudSat-CPR between May 2010 and November 2017 (broken black line marks the median, dotted black lines mark the interquartile range)"*.

The particular statement on P10 L317 could be supported by either CloudSat-CPR line. The statement was revised to clarify that we support it using the CloudSat-CPR observations rather than forward simulations.

*"Thus, we would expect that the CloudSat-CPR, with its -27dBZ MDS (observed performance depicted by the broken black line on Fig. 5b), should have the capability to detect at best 80% of all cloud and/or echoes forming at any given height, de facto missing at least 20% of hydrometeor echoes"*

**P11, L373: should read "...a factor of 0.5 times the pulse length..."**

This typo was corrected.

**P13, L436: should be Fig. 6c**

We would like to thank the reviewer for catching this oversight. It was corrected.

**P14, L474–3: should be something like "...warm marine boundary layer (WMBL) clouds and precipitation, and spaceborne radars' ability to characterize them, is..."**

This typo was corrected.

**P14, L485: should be "...such that..."**

This typo was corrected.

**P14, L491: remove "both"**

The word "both" was removed.

**P15, L505: "...length of its highly sensitive pulse..."**

This typo was corrected.

**P16, L546: remove "study"**

The word "study" was removed.

**P17, L583: should be Fig. 5b**

We would like to thank the reviewer for catching this oversight. It was corrected.

**P17, L589, should be "...this secondary lobe is confined..."**

The word "lobe" was added.

**P22, L728, Figure 1: should be "...ground-track taken in ~7 minutes is shown..."**

We would like to thank the reviewer for catching this oversight. It was corrected.

**Figure 1: I (a color-blind reader) have a lot of difficulty distinguishing the blue dots in Figure 5b from the underlying radar reflectivity (also blue). Since they are not on the same subplot, would it be acceptable to make these dots black as well?**

While the authors understand that it may be difficult for some to decipher the blue dots from their background in Fig. 1b, the authors are hesitant to make them black as this would disrupt the color story used throughout the manuscript. Throughout the manuscript blue is used to represent all things related to the CloudSat-CPR and black is used to represent all things related to KAZR. Since the dots in Fig. 1b are there more so for additional guidance than to present a result, we propose instead to add clarifications about their location in the caption. Hopefully this change and the arrows and labels already included in the figure provide sufficient guidance.

[Figure]

*"Figure 1. Hydrometeor radar reflectivity measured on Feb. 27, 2016 a) by the KAZR located at the Eastern North Atlantic (ENA) observatory over the course of 24 hours and b) by the CloudSat-CPR when it overpassed the 200-km radius region around the KAZR between 15:05:21 and*

*15:06:07 UTC. In (a) the blue line marks the time when CloudSat overpassed KAZR, the red dots represent the location of the ceilometer-determined cloud base and black dots represent the boundaries of the KAZR radar echo; the latter coincides with the center of the first and last radar range gates containing signal (post-processing).* *In (b) blue dots represent the boundaries of the CloudSat-CPR radar echo; they coincide with the center of the first and last radar range gates containing signal (post-processing).* *Also plotted are the CloudSat radar reflectivity c) raw, d) for significant returns (CPR_mask >5), e) for echoes deemed very weak and stronger (CPR_mask > 6) and f) for echoes deemed weak and stronger (CPR_mask > 20)."*

**Figure 1: To make clear the fact that the KAZR and CPR data are on different time- series, it may be useful to mark the time of the CPR overpass with a vertical line on the KAZR timeseries. This would also aid comparison of the cloud fields at the same time.**

We would like to thank the review for this suggestion. A vertical line was added on the KAZR time-series reflecting when the CloudSat overpassed.

**Figure 2: The y-label "factors of the pulse length" is unclear; the label and the sign convention should make it very clear which is the "leading edge" and the which the "trailing edge" of the radar pulse in the direction of propagation.**

We agree with the reviewer. The figure caption was modified to clearly indicate which part of the range-weighting function represents the leading edge of the forward-simulated pulse.

*"Figure 2. Symmetrical (blue) and asymmetrical (black) range weighting functions for the forward simulated radar architectures detailed in Table 1.* *Negative values are associated with the leading edge of the pulse in the direction of propagation."*

**P24, L745–748, Figure 3: In the text is seemed clear that these values (e.g. hydrometeor cover) are fractions of profiles excluding those containing high, mid- and layered clouds. If so, best to re-state this in the caption.**

We would like to thank the review for this suggestion. The captions of Fig.3 and Fig. 5 have been revised to include this detail.

*"Figure 3. For 103 instances where CloudSat overpassed the 200-km radius region centered on the ENA observatory, a) fraction of observed profiles with cloud or rain (i.e., hydrometeor cover) and b) hydrometeor fraction profile. Both estimated from CloudSat-CPR observations within a 200-km radius of the ENA observatory (blue) and ground based KAZR observations collected within ± 1 hr of the CloudSat overpass (black).* *Fractions are estimated based on the total number of observed profiles excluding those determined to contain high, deep or ice clouds."*

*"Figure 5. From ground based KAZR observations collected between 10/2015 and 02/2018, a) profile of cloud (solid black line) and sub-cloud layer rain (dotted black line) fraction, and the fraction of either cloud (solid red line) or sub-cloud-layer rain (dotted red line) echoes located below a certain height.* *Fractions are estimated based on the total number of observed profiles excluding those determined to contain high, deep or ice clouds. [...]"*

**Figure 4: It may again be useful to show the time of the CloudSat overpass on the KAZR timeseries.**

The revised figure now shows observations collected within +/- 1 hr of the CloudSat overpass centered on the overpass time.

[Figure]

*"Figure 4. For 103 instances where CloudSat overpassed the 200-km radius region centered on the ENA observatory, distribution of a) echo base height, and b) echo top height, estimated from CloudSat-CPR observations within a 200-km radius of the ENA observatory (blue) and ground-based KAZR observation collected within ± 1 hr of the CloudSat overpass (grey). For references are examples of hydrometeor radar reflectivity measured on i) Feb. 11, 2017 and ii) Oct. 24, 2016 by the ground based KAZR within ± 1 hr of the CloudSat overpass and by the CloudSat-CPR within 200-km of the KAZR location. Dots on these figures represent the boundaries of the radar echo (black and blue dots for the KAZR and the CloudSat-CPR respectively) and the location of the ceilometer-determined cloud base (red dots)."*

**P26, L 766, Figure 5: should be "...located below a certain height."**

We would like to thank the reviewer for catching this oversight. It was corrected.

**P28, L795, Figure 7: "...which is CloudSat operating with..."**

We would like to thank the reviewer for catching this oversight. It was corrected.

---

## Author Comment (AC2) · 3 Apr 2020

The authors would like to thank all reviewers for their kind words and feedback. A point-by-point response to the reviewer's comments is provided below.

**Reviewer 2**

**I have only one comment, in the statistics of hydrometeor layer properties estimated for days where CloudSat overpassed within 200 km of the ENA station, 4 hrs KAZR and ceilometer observations around the overpass are taken into consideration (Figures 3 and 4). Why do the authors use such a wide time window for their comparison when for cloud-comparison purposes, a length scale of a few tens of kilometers and a time scale of a few minutes is generally acceptable (e.g. Blanchard et al., 2014)? This question is more puzzling in the discussion of the limitations of CloudSat observations, highlighted in Figure 4, with cloud observations up to 1:30 hour time difference with the time of the overpass. I suggest that the authors use a smaller time window for the evaluation of CloudSat performance with KAZAR measurements and provide a justification for the use of this time window and the consequences on the homogeneity of the scene. Similarly, in the discussion of the differences of the statistics observed, it would be good if the examples/arrows pointing to the different CloudSat underestimations/limitations are given in cases that these limitations are visible in the clouds captured from CloudSat and KAZAR collocated cloud observations.**

Following the reviewer's suggestion, the authors modified their intercomparison time window by reducing it from $\pm2$ hrs to $\pm1$ hr around the overpass. The region size and time period used now match those of Protat et al. [2009] which we now cite in the revised version of the manuscript. A mention that this methodology is based on a compromise between keeping the domain size small enough to maintain its homogeneity and capturing a number of cases large enough to reach statistical significance was also added to the revised manuscript.

*"To illustrate how the aforementioned example is representative of the general picture of the WMBL cloud regimes at the ENA, we also compared statistics of hydrometeor layer properties estimated for all instances where CloudSat overpassed within 200 km of the ENA and boundary-layer clouds were the dominant cloud type (Fig. 3 and 4; 103 out of the 138 overpasses). For this comparison, only KAZR and ceilometer observations taken within $\pm1$ hr of the overpass are considered. The predominance of boundary layer clouds is established using KAZR observations taken within $\pm1$ hr of the overpass time. Instances with less than 30% (in time) high or cold clouds are deemed dominated by boundary layer clouds; high or cold clouds present in these instances (if any) are filtered out of the analysis. This region size (for the spaceborne observations) and time period (for the ground-based observation) were selected to match those of Protat et al. [2009] and constitute a compromise between keeping the domain size small enough to maintain its homogeneity (~ 99% ocean by area) and capturing a number of cases large enough to reach statistical significance (103 overpasses)."*

The authors are happy to report that this modified methodology produces results still supporting their initial conclusions. Slight adjustment were made throughout the text to match the revised numbers. For example:

*"Although not expected to perfectly match, the large hydrometeor cover discrepancy between the KAZR (48.1%) and CloudSat-CPR (27.2%) suggest that the CloudSat-CPR fails to detect clouds in more than a few (on the order of ~40%) of the atmospheric columns it samples (Fig. 3a)."*

[Figure]

***Figure 3.*** *For 103 instances where CloudSat overpassed the 200-km radius region centered on the ENA observatory, a) fraction of observed profiles with cloud or rain (i.e., hydrometeor cover) and b) hydrometeor fraction profile. Both estimated from CloudSat-CPR observations within a 200-km radius of the ENA observatory (blue) and ground based KAZR observations collected within ± 1 hr of the CloudSat overpass (black). Fractions are estimated based on the total number of observed profiles excluding those determined to contain high, deep or ice clouds.*

As well as for example:

*"2) The distribution of KAZR-detected cloud top heights also shows the presence of cloud top modes near 1.2 km and frequent occurrences near 2.2 km that are only partially detected by the CloudSat-CPR (Fig. 4a). These elevated cloud tops modes are likely related to the several echo bases between 1.5 and 2.0 km that nearly all went undetected by the CloudSat-CPR (Fig. 4b)."*

[Figure]

*Figure 4. For 103 instances where CloudSat overpassed the 200-km radius region centered on the ENA observatory, distribution of a) echo base height, and b) echo top height, estimated from CloudSat-CPR observations within a 200-km radius of the ENA observatory (blue) and ground-based KAZR observation collected within ± 1 hr of the CloudSat overpass (grey). For references are examples of hydrometeor radar reflectivity measured on i) Feb. 11, 2017 and ii) Oct. 24, 2016 by the ground based KAZR within ± 1 hr of the CloudSat overpass and by the CloudSat-CPR within 200-km of the KAZR location. Dots on these figures represent the boundaries of the radar echo (black and blue dots for the KAZR and the CloudSat-CPR respectively) and the location of the ceilometer-determined cloud base (red dots).*

**The rest of my specific comments are only to encourage more clarity in the presentation of the results or technical corrections.**

**1. Page 4, line 313 – 326: Although mentioned in the legend of Figure 5b, the CloudSat blue line in fig. 5b is not mention in the paragraph.**

We would like to respectively point out that the CloudSat blue line is mentioned in the appendix.

*"The gaussian range weighting function depicted in Fig. 2 produces a forward-simulated surface echo return similar, in intensity and vertical extent, to the surface echo observed by the CloudSat-CPR under clear sky conditions (compare the royal blue line and black lines in Fig. 5b)."*

**2. Page 11, line 391: There is a typo in the factor.**

The sentence was revised.

*"The vertical stretching of cloud tops results from additional power being focused between a factor of 0.0 and 0.5 times the pulse length on the leading edge of the pulse (comparing the range-weighting function of EarthCARE-CPR to that of the CloudSat-CPR; respectively the black and blue line on Fig. 2)."*

**3. Page 15, line 510: Apart from a ceilometer, the synergy with the EarthCARE lidar (ATLID) could help correct the cloud top height.**

We would like to thank the review from this recommendation. It was added into the text.

*"Synergy with the collocated Atmospheric Lidar (ATLID) could potentially help correct cloud top height, however, such corrections would only be possible in single layer conditions and alternative techniques would need to be developed to improve the EarthCARE-CPR's ability to accurately estimate the vertical extent of multi-layer boundary layer clouds."*

**Reference:**

**Blanchard, Y., J. Pelon, E.W. Eloranta, K.P. Moran, J. Delanoë, and G. Sèze, 2014: A Synergistic Analysis of Cloud Cover and Vertical Distribution from A- Train and Ground-Based Sensors over the High Arctic Station Eureka from 2006 to 2010. J. Appl. Meteor. Climatol., 53, 2553–2570, https://doi.org/10.1175/JAMC-D-14- 0021.1.**

Protat, A., D. Bouniol, J. Delanoë, E. O'Connor, P. May, A. Plana-Fattori, A. Hasson, U. Görsdorf, and A. Heymsfield (2009), Assessment of CloudSat reflectivity measurements and ice cloud properties using ground-based and airborne cloud radar observations, *Journal of Atmospheric and Oceanic Technology*, *26*(9), 1717-1741.

---

## Author Comment (AC3) · 3 Apr 2020

The authors would like to thank the reviewer for his/her kind words and feedback.